# Mother-to-Child Transmission of Hepatitis B Virus in Ethiopia

**DOI:** 10.3390/vaccines9050430

**Published:** 2021-04-26

**Authors:** Asgeir Johannessen, Bitsatab Mekasha, Hailemichael Desalegn, Hanna Aberra, Kathrine Stene-Johansen, Nega Berhe

**Affiliations:** 1Department of Infectious Diseases, Vestfold Hospital Trust, 3103 Tønsberg, Norway; 2Faculty of Medicine, Institute for Clinical Medicine, University of Oslo, 0315 Oslo, Norway; 3Regional Centre for Imported and Tropical Diseases, Ullevål, Oslo University Hospital, 0424 Oslo, Norway; nega.berhe@aau.edu.et; 4Medical Department, St. Paul’s Hospital Millennium Medical College, 1230 Addis Ababa, Ethiopia; bitsatabm@gmail.com (B.M.); hailemichael.desalegn@sphmmc.edu.et (H.D.); hanna.aberra@aau.edu.et (H.A.); 5Department of Virology, Norwegian Institute of Public Health, 0456 Oslo, Norway; Kathrine.Stene-Johansen@fhi.no; 6Aklilu Lemma Institute of Pathobiology, University of Addis Ababa, 1230 Addis Ababa, Ethiopia

**Keywords:** viral hepatitis, resource-limited settings, Africa, transmission, pregnancy, pediatric

## Abstract

High viral load and positive hepatitis B e-antigen (HBeAg) results are risk factors for mother-to-child transmission (MTCT) of hepatitis B virus (HBV). In sub-Saharan Africa, little is known about the distribution of these risk factors, as well as early childhood HBV transmission. In this study, Ethiopian women aged 18–45 years with chronic hepatitis B were assessed for the presence of HBeAg and high viral load. Their children below 4 years of age were invited for assessment of viral markers, defining active HBV infection as a positive hepatitis B s-antigen (HBsAg) and/or detectable HBV DNA. In total, 61 of 428 HBV-infected women (14.3%) had a positive HBeAg result and/or a high viral load. Of note, 26 of 49 women (53.1%) with viral load above 200,000 IU/mL were HBeAg negative. Among 89 children born of HBV-infected mothers (median age 20 months), 9 (10.1%) had evidence of active HBV infection. In conclusion, one in seven women with chronic hepatitis B had risk factors for MTCT, and HBeAg was a poor predictor of high viral load. One in ten children born of HBV-infected women acquired HBV-infection despite completing their scheduled HBV vaccination at 6, 10 and 14 weeks of age.

## 1. Introduction

Hepatitis B virus (HBV) infection is common throughout the world and it is estimated that 257 million people are living with chronic HBV infection globally [1]. In 2016, the World Health Organization endorsed an ambitious plan to eliminate viral hepatitis as a public health threat by 2030, aiming to reduce new infections by 90% and mortality by 65% [2]. To reach this target, a number of interventions are needed, including scaling up coverage of infant vaccination to a minimum 90% of infants, birth dose vaccination to a minimum 90% of neonates within 24 h of birth, diagnosis of 90% of people infected with HBV, and antiviral treatment to a minimum 80% of people who are diagnosed and eligible for treatment [2,3]. However, global progress to reach the elimination goals has been slow, particularly in sub-Saharan Africa where the burden of hepatitis B is high [4].

The risk of developing chronic HBV infection depends on the age at which HBV exposure occurs; 90% of newborns infected with HBV will develop chronic hepatitis B (CHB), compared to less than 5% of those infected as adults [5]. Of those who develop CHB, an estimated 20% will later in life develop cirrhosis, liver failure and/or hepatocellular carcinoma, which translates to nearly 900,000 HBV-related deaths globally each year [1,6]. This is quite comparable to death tolls from other major infectious diseases such as human immunodeficiency virus (HIV), tuberculosis and malaria, which traditionally have received far more attention and funding [7]. Indeed, unlike most communicable diseases, the absolute burden and relative rank of viral hepatitis has increased steadily from 1990 till now [8].

The health consequences of being infected with HBV early in life are dire, and mother-to-child transmission (MTCT) contributes significantly to the persistence of the high numbers of HBV carriers globally [9]. Perinatal transmission is strongly associated with a positive hepatitis B e-antigen (HBeAg) result and/or a high HBV DNA viral load of childbearing women [10,11,12,13]; indeed, up to 90% of babies born to HBeAg positive mothers become chronic HBV carriers compared to 5%–31% of babies born to HBeAg negative mothers [14].

An effective HBV vaccine has been available since the 1980s, which in studies from Asia has been shown to reduce the incidence of MTCT by 90% when the initial dose is given immediately after birth [15,16]. The addition of hepatitis B immune globulin (HBIG) immediately after birth can further reduce the risk of HBV infection to less than 5% [17]. In many resource-limited countries, however, the initial dose is given as a pentavalent vaccine in the EPI (Expanded Program on Immunization) program at 6 weeks of age, and consequently the neonate is not protected against perinatal transmission. Indeed, less than 10% of infants in sub-Saharan Africa received the birth dose HBV vaccine in 2015 [1].

To date, there is a paucity of data on transmission of HBV in Africa. The present study, which was nested in a large HBV cohort in Ethiopia and included treatment-naïve HBV-infected women of childbearing age, aimed to study: (a) the prevalence of risk factors for MTCT (positive HBeAg status and high HBV DNA viral load), (b) the association between HBeAg and HBV DNA viral load, and (c) vertical HBV transmission. This type of real-life data from sub-Saharan Africa is vital to design targeted public health interventions to combat viral hepatitis on the continent.

## 2. Materials and Methods

### 2.1. Study Setting and Participants

Ethiopia is a low-income country in east Africa with an estimated hepatitis B s-antigen (HBsAg) prevalence of 9.4% [18]. The HBV vaccine was included in the national immunization program in 2007, given as a pentavalent vaccine (together with diptheria, tetanus, pertussis and haemophilus influenzae type b) at 6, 10 and 14 weeks of age. Although monovalent HBV vaccine is available at local pharmacies, it is expensive and not routinely given to children born of HBsAg positive mothers.

In 2015, our group established a treatment centre for CHB at St. Paul’s Hospital Millennium Medical College in Addis Ababa. A total of 1303 adults aged 18 or older with chronic hepatitis B were enrolled and have been followed up since. Chronic hepatitis B was defined as a positive HBsAg rapid test for more than 6 months [19]. Individuals with HIV infection were not enrolled, but rather referred to the nearest HIV care and treatment center. The patients were a mixture of symptomatic individuals seeking medical attention for overt liver disease and asymptomatic HBsAg positive patients screened at antenatal clinics, blood banks, etc. The clinical and operational aspects of this cohort have been published previously [20,21,22,23].

In the present study we first included HIV-negative HBsAg-positive women of childbearing age (18–45 years) to assess risk factors for MTCT and the association between HBeAg and HBV DNA viral load. Subsequently, women who were pregnant or lactating at inclusion, or became pregnant during the first year of follow-up, were invited to have their offspring below 4 years of age tested for markers of hepatitis B infection. The vaccination cards were used to collect data on the infants’ vaccination status. However, since the vaccination cards are first dispensed at the routine vaccination visit at 6 weeks’ age, information about the birth dose HBV vaccine and HBIG was collected verbally from the mothers.

The study was approved by the National Research Ethics Review Committee (Ref. No.: 3.10/829/07) in Ethiopia and by the Regional Committees for Medical and Health Research Ethics (Ref. No.: 2014/1146) in Norway. The study was conducted in accordance with the Declaration of Helsinki. Written informed consent was obtained from all study subjects.

### 2.2. Laboratory Investigations

Liver function tests (Humalyzer 3000, HUMAN, Wiesbaden, Germany), serology (HBsAg/anti-HBs/anti-HBc, Elisys Uno, HUMAN, Wiesbaden, Germany), HBV DNA viral load (rt2000 real-time PCR, Abbott Molecular, Des Moines, IL, USA), and HBeAg (VIDAS HBe/anti-HBe, BioMerieux, Marcy-l’Étoile, France) were measured in all women at inclusion in the program. Transient elastography (Fibroscan 402, Echosens, Paris, France) was performed by trained staff after at least two hours of fasting. The median of 10 readings was employed and the result was discarded if the interquartile range (IQR) divided by the median exceeded 30%, as recommended by the manufacturer. For safety reasons we avoided liver stiffness measurements during pregnancy; instead, the nearest result before or after birth was used.

The children were tested for viral markers (HBsAg, anti-HBc, anti-HBs, HBV DNA) using the same assays and kits as given above. A positive HBsAg result and/or HBV DNA in a child below 12 months of age was taken as proof of vertical HBV transmission. A positive HBsAg result and/or HBV DNA in a child older than 12 months was taken as proof of active HBV infection, either horizontal or vertical. A positive anti-HBc result without HBsAg in a child older than 24 months was considered proof of past infection with HBV; presence of anti-HBc in younger children was not considered since it may represent persistence of maternal antibodies transferred in utero [24].

### 2.3. Statistical Analyses

Baseline characteristics were summarized using descriptive statistics. Continuous variables were compared using Mann–Whitney U-tests and categorical variables were compared using Chi square tests. Logistic regression models were used to identify factors associated with active HBV infection. SPSS version 26.0 software (SPSS Inc., Chicago, IL, USA) was used to analyze the data. The level of significance was set at *p* < 0.05.

## 3. Results

### 3.1. Women of Childbearing Age

Out of 1303 HBsAg positive patients in the cohort, 428 HIV negative women between 18 and 45 years of age with a full virological assessment at baseline were included in this analysis. The median age was 29 years (interquartile range [IQR] 25–35). Overall, 49 (11.4%) women had a HBV DNA viral load >200,000 (>5.3 log_10_) IU/mL and 35 (8.2%) had a positive HBeAg result at baseline. In total, 61 of 428 women (14.3%) had a high-risk profile for vertical transmission of HBV, either due to a positive HBeAg result and/or a high HBV DNA viral load.

HBeAg positive women were significantly younger and had higher AST/ALT levels compared to HBeAg negative women; however, there was no significant difference in liver stiffness at baseline (Table 1).

Women who were HBeAg positive had on average a higher HBV DNA viral load compared to HBeAg negative women (Figure 1). Still, 26 of 49 women (53.1%) with HBV DNA >200,000 (>5.3 log_10_) IU/mL were HBeAg negative (Table 1).

### 3.2. Pregnant and Lactating Women

One-hundred-and-twenty-three women were pregnant (*n* = 64) or lactating (*n* = 59) at inclusion, and another 13 became pregnant during the first year of follow-up. The median age was 28 years (IQR 25–30). Median (IQR) ALT, AST, HBV DNA viral load and liver stiffness was 18 U/L (15–24), 20 U/L (17–25), 508 IU/mL (127–3706) and 4.5 kPa (3.7–5.4), respectively. Nine (6.6%) women had an HBV DNA viral load >200,000 IU/mL, 4 of whom were HBeAg positive and 5 HBeAg negative. Overall, 9 (6.6%) women were HBeAg positive. Only three women received antiviral therapy in pregnancy, all of whom were among those who became pregnant after enrollment in the CHB program.

When restricting the analysis to only women who were pregnant at inclusion (*n* = 64), there was a higher proportion of HBeAg positive women who had a HBV DNA viral load >200,000 IU/mL (Table 2); however, numbers were small and did not reach statistical significance (*p* = 0.094).

### 3.3. Children Born of HBsAg Positive Women

Out of 136 children who were invited to a clinical evaluation, 89 (65.4%) attended the clinic for assessment. The median age at the time of the assessment was 20 months (IQR 16–29). All the children had taken three doses of HBV vaccine in accordance with the national immunization schedule, whereas 43 (48.3%) reported taking the birth dose HBV vaccine and 55 (65.5%) reported taking HBIG (Table 3).

Nine children (10.1%) had evidence of active HBV infection, either by a positive HBsAg result (*N* = 8) and/or detectable HBV DNA (*N* = 3). These children were aged 15 to 24 months and hence none were confirmed to have had disease transmitted vertically. Key characteristics of these nine children and their mothers are depicted in Table 4.

None of 17 children below 12 months of age had evidence of active HBV infection. Only 1 of 26 children (3.8%) above 24 months of age had a positive anti-HBc result, indicating past HBV infection. No statistically significant predictors of active HBV infection could be identified (Table 5).

Sixty-three children (70.8%) had positive anti-HBs results, indicating vaccine response. There was a numerically higher proportion with vaccine response in the group who received the birth dose HBV vaccine (79.1 vs. 63.4%); however, the difference did not reach statistical significance (*p* = 0.112). There was no significant difference in the detection of anti-HBs in children below and above 24 months of age (66.7% vs. 76.3%; *p* = 0.322).

## 4. Discussion

About one of seven women (14.3%) of childbearing age had risk factors for MTCT, viz. a high HBV DNA viral load and/or a positive HBeAg result. These women are at high risk of transmitting HBV infection to their offspring in the absence of giving the birth dose HBV vaccine to the infant and/or antiviral therapy during pregnancy [15,25]. Data from other African countries are scarce, but a study of 224 HBsAg positive childbearing women in Togo found that 9.0% had HBV DNA >1,000,000 IU/mL [26]. Moreover, a study from South Africa found that 17.1% of HIV negative HBsAg positive pregnant women were HBeAg positive [27], whereas a study from Cameroon found that 22.7% of 259 pregnant women were HBeAg positive [28]. A recent meta-analysis estimated that there are 367,250 cases of perinatal HBV transmission per year in sub-Saharan Africa [29]. Thus, the risk of MTCT of HBV in Africa is not negligible, and identification of high-risk women is crucial to implement preventive measures.

The use of HBeAg as a proxy for high HBV DNA viral load has been suggested in settings where HBV DNA measurements are unavailable [30]. Our data suggest that this approach lacks sensitivity in Ethiopian patients; indeed, less than half of the mothers with HBV DNA viral load above 200,000 IU/mL were HBeAg positive and none of the HBV-infected children had mothers with high HBV DNA viral load. Similar results have been reported in previous studies from other African countries [28,31]. On the other hand, a recent meta-analysis by Boucheron and colleagues found that the pooled sensitivity of HBeAg testing to identify HBV DNA above 200,000 IU/mL in pregnant women was 88.2% [30]. It is unclear why our data differed from the study by Boucheron, but it should be noted that the meta-analysis mainly comprised Asian patients. However, in a subgroup analysis restricted to African studies only, Boucheron and colleagues found that the sensitivity was still high (82.0%). Age is another possible explanation for the poor correlation between HBeAg and viral load in our study, since HBeAg is lost over time in HBV-infected women [32]. In our study we did not include individuals younger than 18 years, although teenage pregnancies are common in sub-Saharan Africa. Indeed, the median age in our study was 29 years, and Boucheron and colleagues reported a lower sensitivity in studies with a median maternal age of 28 years or older [30]. Clearly, more studies are needed to assess whether the use of HBeAg as an indicator for high HBV DNA viral load is valid in African women.

All the children in this study received HBV vaccine (three doses) in accordance with the national immunization guidelines, and a significant proportion also reported the use of the birth dose HBV vaccine and/or HBIG. In contrast to international recommendations [5], there were more women who reported giving HBIG than the birth dose HBV vaccine to their neonates, even though HBIG is approximately ten times more expensive (100 vs. 10 USD in Ethiopia). It is possible that the more expensive remedy was perceived as the better option, and clearly there is a need for improved health education at antenatal clinics to make sure that priority is given to the cheaper and more effective remedies.

Despite excellent vaccination coverage, as many as one in ten children had evidence of HBV infection in our study. Among these HBV infected children, more than half received the birth dose HBV vaccine and/or HBIG, indicating that the vaccination is less effective in this setting. This is disturbing since HBV infection acquired perinatally or in early childhood is likely to lead to chronicity. Indeed, data from the Gambia have shown that two-thirds of adults who require antiviral therapy were infected as infants [33]. Recent estimates of vaccine coverage in sub-Saharan Africa indicate that 80% of children receive three doses of HBV vaccine, but only 10% receive the birth dose vaccine and 0% get HBIG [34]. The same study estimated an alarmingly high HBsAg prevalence of 3.4% among children aged 5 years in the African region, compared to 0.5% in the Western Pacific region and 0.1% in the European region. Clearly, prevention of vertical and early horizontal HBV transmission will be key to reducing HBV-related mortality on the African continent in the decades to come.

Only about two-thirds of the children in our study had detectable anti-HBs indicative of vaccine response and protection against HBV infection, despite full vaccine coverage. A study from Tanzania found protective antibodies in 69.3% of vaccinated children below 5 years of age [35]. Similarly, a study from Mozambique found that 76% were protected at a median age of 5 years [36]. These levels are lower than those reported from Asian and European countries where protective antibodies are typically found in more than 90% of vaccinees [37], and further studies on vaccine efficacy and factors associated with vaccine failure are warranted in sub-Saharan Africa.

Our study had some limitations. First, this was a cross-sectional study and we were unable to ascertain when the actual HBV infection had taken place. In other words, we could not ascertain that mother-to-child transmission had indeed occurred. However, horizontal HBV transmission is assumed to be rare in children below 2 years of age [38], and all the HBV positive children in this study were younger than 24 months. Moreover, horizontal transmission is extremely rare in children who completed a series of three doses of infant HBV vaccine [39]. Thus, it can be assumed that the majority of the HBV infected children in the present study were infected vertically. Second, maternal HBeAg status and HBV DNA viral load during pregnancy was not available in women who were included when they were breastfeeding. Hence, some of these women could have undergone HBeAg seroconversion between the pregnancy and enrollment. However, this could be assumed to be a rare event. Third, only about two-thirds of the invited children were eventually assessed for viral markers. We cannot exclude a selection bias, although it seems unlikely. Fourth, the study participants were recruited from an ongoing urban hospital-based CHB cohort, and were therefore not representative of the wider community. Indeed, women who became pregnant in this program were educated about the importance of the birth dose vaccine and HBIG. This explains the widespread use of these commodities, which are beyond reach for the vast majority of Ethiopian women.

In summary, we found that one in seven HBV-infected women of childbearing age had risk factors for mother-to-child transmission of HBV, viz. a high HBV DNA viral load or positive HBeAg result. Reliance on HBeAg positivity alone would fail to detect more than half of the patients with high HBV DNA viral load, thereby missing a substantial proportion of the high-risk pregnancies. Despite excellent adherence to the national immunization program, as many as 10% of the children examined had already acquired HBV infection, raising concerns about the effectiveness of the current vaccination strategy.

## Figures and Tables

**Figure 1 vaccines-09-00430-f001:**
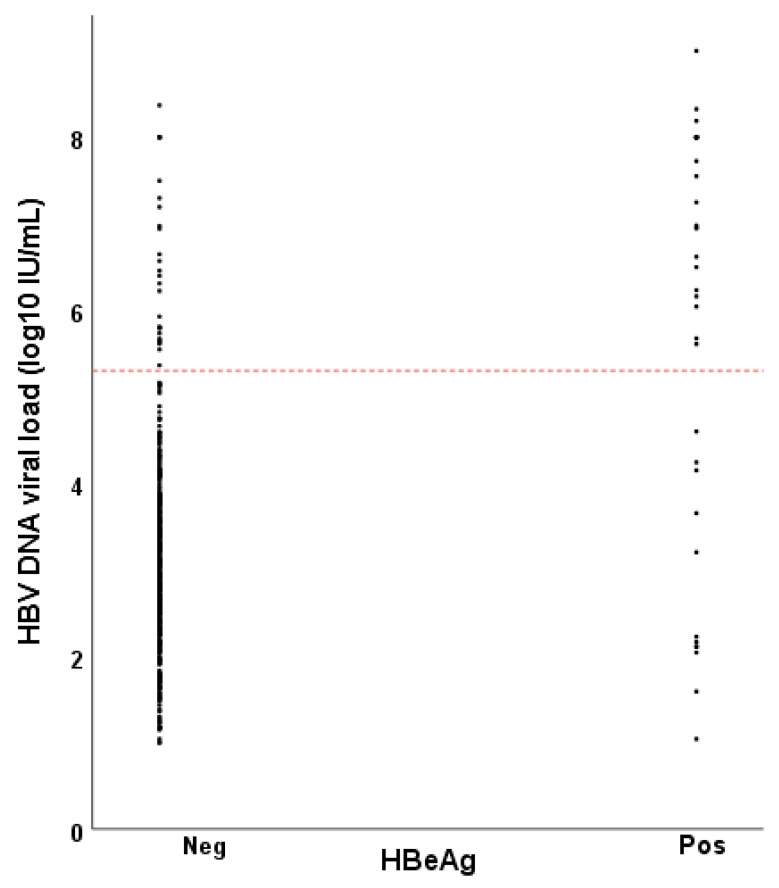
Hepatitis B virus (HBV) DNA viral load as a function of HBeAg status among women of childbearing age with chronic hepatitis B in Ethiopia. The reference line at 5.3 log_10_ IU/mL (200,000 IU/mL) indicates the threshold for high risk of vertical transmission of hepatitis B virus.

**Table 1 vaccines-09-00430-t001:** Baseline characteristics of women of childbearing age with chronic hepatitis B virus infection in Addis Ababa, Ethiopia.

Characteristics	HBeAg Negative Median (IQR)/N (%) *N* = 393	HBeAg Positive Median (IQR)/N (%) *N* = 35	*p*
Age (years)	29 (25–35)	23 (19–29)	0.006
Alanine aminotransferase (U/L)	20 (16–27)	26 (17–48)	0.020
Aspartate aminotransferase (U/L)	22 (19–27)	27 (19–43)	0.015
Liver stiffness (kPa)	4.9 (4.1–6.1)	6.3 (4.5–7.7)	0.421
HBV DNA viral load (log_10_ IU/mL)	3.1 (2.3–3.8)	6.5 (3.7–8.0)	<0.001
<5.3 log_10_ IU/mL	367 (93.4)	12 (34.3)	<0.001
>5.3 log_10_ IU/mL	26 (6.6)	23 (65.7)	

**Table 2 vaccines-09-00430-t002:** HBV DNA viral load as a function of HBeAg in pregnant women with hepatitis B virus infection.

HBV DNA Viral Load	HBeAg Negative *N* (%)	HBeAg Positive *N* (%)	Total
<5.3 log_10_ IU/mL	59 (98.3)	2 (66.7)	61
>5.3 log_10_ IU/mL	1 (1.7)	1 (33.3)	2
Total	60	3	63

Missing HBeAg in one woman.

**Table 3 vaccines-09-00430-t003:** Immunization status and HBV infection of 89 children born to HBsAg positive mothers in Ethiopia.

	HBV Vaccine	HBV Vaccine + HBIG	HBV Vaccine + Birth Dose	HBV Vaccine + Birth Dose + HBIG
*N*	13	33	21	22
Antiviral therapy in pregnancy	0	3	0	0
HBV infected (%)	2/13 (15.4)	3/33 (9.1)	1/21 (4.8)	3/22 (13.6)

**Table 4 vaccines-09-00430-t004:** Characteristics of nine children with active HBV infection.

	Variable	1	2	3	4	5	6	7	8	9
Child	Birth dose vaccine within 24 h of delivery	N	Y	Y	Y	Y	N	*	N	N
	HBIG within 24 h of delivery	N	N	Y	Y	Y	Y	Y	N	Y
	3 doses of vaccine at 6, 10 and 14 weeks of age	Y	Y	Y	Y	Y	Y	Y	Y	Y
	Age at sample collection (months)	15	16	17	20	20	20	20	23	24
	HBsAg	pos	pos	pos	pos	pos	pos	pos	neg	pos
	HBV DNA (IU/mL)	0	0	0	0	21	0	0	881,000	89
Mother	Age (years)	40	29	24	27	22	39	37	25	28
	Maternal blood sample collection	preg	preg	preg	preg	preg	lact	preg	preg	lact
	HBeAg	neg	neg	neg	neg	neg	neg	neg	neg	neg
	HBV DNA (IU/mL)	3120	193,000	206	53	33	64	3880	456	824
	ALT (U/L)	15	14	24	12	33	23	11	18	41
	Liver stiffness (kPa)	4.2	*	6.8	5.0	4.4	5.3	7.3	*	4.8
	Antiviral therapy in pregnancy	N	N	N	N	N	N	N	N	N

N = no; Y = yes; pos = positive; neg = negative; preg = in pregnancy; lact = during lactation. * Missing data.

**Table 5 vaccines-09-00430-t005:** Factors associated with hepatitis B virus infection in children born to HBsAg positive mothers, Addis Ababa, Ethiopia.

Variable	Total (*N* = 89) *N* (%)	HBV Infected (*N* = 9) *N* (%)	Crude Odds Ratio (95% Confidence Interval)	*p*
Sex				0.507
girl	39 (43.8)	3 (33.3)	0.6 (0.1–2.6)	
boy	50 (56.2)	6 (66.7)	1	
Birth dose HBV vaccine ^1^				0.944
yes	43 (51.2)	4 (50.0)	0.9 (0.2–4.1)	
no	41 (48.8)	4 (50.0)	1	
Hepatitis B immune globulin ^1^				0.937
yes	55 (65.5)	6 (66.7)	1.1 (0.2–4.6)	
no	29 (34.5)	3 (33.3)	1	
HBeAg positive mother ^2^				N.D.
yes	6 (6.8)	0 (0)	*	
no	82 (93.2)	9 (100)		
High-viremic mother (HBV DNA >200,000 IU/mL)				N.D.
yes	6 (6.7)	0 (0)	*	
no	83 (93.3)	9 (100)		
Antiviral therapy in pregnancy				N.D.
yes	3 (3.4)	0 (0)	*	
no	86 (96.6)	9 (100)		

^1^ Missing data in 5 children. ^2^ Missing data in 1 mother. * Could not be calculated. N.D., not determined.

## Data Availability

The data presented in this study are available on request from the corresponding author. The data are not publicly available due to privacy considerations (study involving children).

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
