# Peer review of "Mother-to-Child Transmission of Hepatitis B Virus in Ethiopia"

_vaccines, 2021, doi:10.3390/vaccines9050430_

Round 1

Reviewer 1 Report

Johannessen et al. studied the risk of HBV transmission from Mother-to-Child in Ethiopia.  HBV infected Ethiopian women and their children were assessed for the presence of HBV markers such as HBeAg and high viral load, HBS Ag ,anit-HBc,...etc. 61 out of 428 HBV-infected women had a positive HBeAg and/or a high 18 viral load. The authors also showed that 26 of 49 women (53.1%) with viral load above 200,000 IU/ml were HBeAg nega-19 tive. 9 out 89 children born of HBV-infected mothers had evidence of active HBV infection.

The authors cocnulded that  1/7 of HBV infected women had risk factors for MTCT, and HBeAg was a poor predictor of high viral load. And 1/10 children born of 22 HBV-infected women acquired HBV-infection despite completing their scheduled HBV vaccination  at 6, 10 and 14 weeks of age.

Although the data is important, I got confused from the texted results and the figures and/or tables preseneted. The data is not consistent and this could harmefully impact the statistic and the final interpretaing conclusions.

In details

1- Page 3: line 129. The author mentioned "The median age was 29 years (interquartile range [IQR] 25-35)"

Going to table 1: these finding for HBe Ag negative women, not for the whole cohort. Please calrify

2- page 3 line 130-131. The author mentioned "Overall, 49  (11.4%) women had HBV DNA viral load >200 000 (>5.3 log10) IU/ml and 35 (8.2%) had a  positive HBeAg at baseline"

Going to Figure 1 and table, these data is not accurate. I can see only 20 subjects of HBe Ag negative, abd 17 subjects of HBe Ag positive with a viral load > 200 000 (>5.3 log10) IU/ml.

3- Similalrly in the result section 3.2 and table 2

Please go again to our tables and results,  correct them, and do again the staitisic analysis according to the new calculations.

Author Response

Thanks for your kind review of our manuscript. We have carefully gone over your comments and addressed them point-by-point below:

  1. Median age (p3, line 132): Since >90% of the women were HBeAg positive, the median age in the whole group was identical to the median age of the HBeAg positive women.
  2. Viral load >200 000 IU/ml (p3, line 133): The numbers in the text are indeed correct and consistent with the numbers in Table 1. I assume your question comes from counting dots in Figure 1, but this is not feasible as since some of the dots are overlapping.
  3. Viral load >200 000 IU/ml (p4, line 159): Again, the numbers are correct. In section 3.2 we describe 123 pregnant/lactating women, of whom 9 had viral load >200 000 IU/ml. In section 3.3 with Table 2 we describe 89 children born of HBV-infected mothers, of whom 6 had high-viraemic mothers.

Reviewer 2 Report

This is a straightforward study that presents the HBV data on obtained from a clinic in Addis Araba using approved protocols.

Overall the data are interesting and critically analyzed - importantly, they present a clear picture regarding HBV infection, diagnostic markers and transmission in this cohort. Thus they should prove useful to the field. I only have a few minor suggestions to polish the presentation:

Minor points

  1. Introduction, line 59: change ‘ the initial dose is gives…’ to the initial does is given
  2. Introduction, line 60: define the acronym EPI for the reader.
  3. Results. Line 167: define the acronym HBIG for the reader.
  4. Table 2: I assume that the heading for the fourth column be 95% confidence interval rather than confidence ‘internal’. Also the abbreviation N.S. should be defined in the notes under the table (and since p values of > 0.5 are not considered significant this abbreviation might be useful to add to the entire column – or perhaps ND (not determined) would be more appropriate here?)

Author Response

Thanks for your kind review of our manuscript. We have addressed your comments as requested:

  1. Introduction, line 59, «…the initial dose is given…»: Done.
  2. Introduction, line 60, define EPI: Done.
  3. Results, line 170, define HBIG: It was already defined at p2 line 57.
  4. Table 2 (which is Table 3 in the revised version): «confidence interval» - done. Define «N.D.» – done.

Round 2

Reviewer 1 Report

The authors answered my questions and I do not have any further concerns

Author Response

Thanks again for your kind review.